# Human tonsil organoids reveal innate pathways modulating humoral and cellular responses to ChAdOx1

Maria Fransiska Pudjohartono[1,2,3], Kate Powell[4], Eleanor Barnes[1,4,5], Paul Klenerman[1,2,4,5‡], Nicholas M. Provine[1,2,3‡*]

**1** Translational Gastroenterology and Liver Unit, University of Oxford, Oxford, United Kingdom, **2** Pandemic Sciences Institute, Nuffield Department of Medicine, University of Oxford, Oxford, United Kingdom, **3** Centre for Human Genetics, Nuffield Department of Medicine, University of Oxford, Oxford, United Kingdom, **4** Peter Medawar Building for Pathogen Research, Nuffield Department of Medicine, University of Oxford, Oxford, United Kingdom, **5** NIHR Oxford Biomedical Research Centre, University of Oxford, Oxford, United Kingdom

‡ Jointly led this project.
* nicholas.provine@ndm.ox.ac.uk

## Abstract

The COVID-19 pandemic response demonstrated the effectiveness of adenovirus vector vaccines in inducing protective cellular and antibody responses. However, we still lack mechanistic understanding of the factors regulating immunity induced by this platform, especially innate pathways. We utilized a human tonsil organoid model to study the regulation of adaptive responses to ChAdOx1 nCoV-19. Innate activation and cytokine release occurred within 24 hours and T and B cell activation and antigen-specific antibody secretion occurred during the ensuing 14-day culture. Among the immune cell populations, plasmacytoid dendritic cells (pDCs) exhibited the highest ChAdOx1 transduction levels. pDC-derived IFN-α was critical for humoral responses, but production of antigen in pDCs was dispensable. Furthermore, IL-6 enhanced humoral responses in both IFN-α-dependent and independent manners, indicating intricate signaling interplay. IFN-α and IL-6 also regulated the function of vaccine-activated CD4$^+$ T cells, including T$_{FH}$. These data provide key insights into innate pathways regulating ChAdOx1-induced immunity and highlights the promise of this model for vaccine platform mechanistic studies.

## Author summary

Despite the success of adenovirus-based vaccines during the COVID-19 pandemic, we still lack understanding of the factors dictating the responses to these vaccines. As vaccine responses are generated in secondary lymphoid organs (SLOs), it is critical to study the processes happening in human SLOs. Tonsillectomies provide a source of human SLO tissue for vaccine studies. We used a

provided the original author and source are credited.

**Data availability statement:** All relevant data are within the manuscript and its Supporting Information files.

**Funding:** M.F.P. is a recipient of the Jardine Foundation Scholarship. E.B. is supported as an NIHR Senior Investigator and the NIHR Oxford Biomedical Research Centre. P.K. is supported by a Wellcome Senior Fellowship [222426/Z/21/Z], the NIH (U19 I082360), the NIHR Oxford Biomedical Research Centre, an NIHR Senior Fellowship, and the University of Oxford NDM COVID-19 emergency relief fund. N.M.P. is supported by a Pandemic Sciences Institute career fellowship and a Wellcome Career Development Award [227217/Z/23/Z]. This work is supported by the UKRI MRC IMMPROVE consortium (MR/Y004450/1). The views expressed are those of the author(s) and not necessarily those of the NHS, the NIHR, or the Department of Health. The funders had no role in study design, data collection, analysis, decision to publish, or preparation of the manuscript.

**Competing interests:** I have read the journal's policy and the authors of this manuscript have the following competing interests: E.B. consults for AstraZeneca, Roche and Vaccitech and has patents in ChAdOx1 HBV and HCV vaccines. P.K. has received consulting fees from UCB, Biomunex, AstraZeneca and Infinitopes. N.M.P. has received consulting fees from Infinitopes.

human tonsil organoid model to study the factors regulating immune responses to ChAdOx1 vaccines. The model recapitulated ChAdOx1-induced antigen-specific antibody production and white blood cell responses over 14 days of culture. The model also showed innate immune activation at 24 hours, measured by expression of vaccine products, innate activation markers on all immune cell types, and cytokine production. Based on vaccine product expression and cytokine production patterns, we identified plasmacytoid dendritic cells (pDCs) as an important regulator of multiple aspects of ChAdOx1-induced responses, primarily through production of the cytokine interferon-α (IFN-α). Another cytokine, interleukin-6 (IL-6), also modulated antibody and cellular responses in both IFN-α-dependent and independent manners. Overall, we utilized the human tonsil organoid model to reveal a critical pDC–IFN-α–IL-6 signaling axis in regulating ChAdOx1-induced immune responses, thus illustrating the potential of this model to further investigate this vaccine technology.

## Introduction

The COVID-19 pandemic response expedited the implementation of new vaccine platforms, including adenovirus vector vaccines. Several adenovirus vector vaccines were introduced against SARS-CoV-2, with effectiveness in inducing antibodies to SARS-CoV-2 and protecting from COVID-19 disease [1–3]. However, despite the successful deployment of these vaccines, mechanistic understanding of the factors governing the humoral responses to adenovirus vector vaccines is still lacking, with most studies focused on using the technology to induce cellular immunity [4]. Thus, there is a major gap in our understanding of this vaccine platform.

Beyond a few select studies, little is known about the regulation of humoral immunity to the transgene antigen cargo of adenoviral vector vaccines [4]. Protection against COVID-19 is associated with the production of IgG antibodies targeting SARS-CoV-2, which requires the development of antigen-specific plasma cells and memory B cells in germinal centers (GC) in lymph nodes. In the GC reaction, CD4+ T cells and particularly follicular helper T cells ($T_{FH}$) play an important role in providing costimulatory signaling to B cells. The induction of antibodies towards the transgene of Ad vaccines is known to be primarily driven by GCs [5], requiring CD4 T cells [6,7] and specifically $T_{FH}$ [5].

However, less is known about the innate signals that initiate the humoral responses to adenoviral vaccines. Innate immune responses can modulate the GC response and outcomes, with innate immune phenotypes being associated with adaptive immune response outcomes to other vaccine platforms [8,9]. For adenoviral vaccines, TLR4 signaling has been shown to promote antibody responses [10]. One study has found that induction of antibodies against the adenovirus viral particle itself is dependent on type I interferon, which regulated multiple steps of B cell differentiation [11]. However, it is unclear if similar signals regulate the immune response to the transgene antigen. A recent study in NHPs found a correlation between type I

IFN levels and antibody titers against a SARS-CoV-2 spike transgene [12], suggesting an association. Direct experimental validation is required, and more generally, a broad survey of critical innate immune pathways is needed to unravel the biology of these vaccines.

To investigate the role of these innate-adaptive immune cell interactions in modulating humoral responses, it is critical to model these processes occurring at the site of vaccine responses – secondary lymphoid tissues. A recently described in vitro human lymphoid tissue organoid model utilizing tonsil tissue has shown utility for investigating the regulation of human B cell responses [13–16]. Providing a model for studying both antibody and cellular biology in an antigen-specific manner, the system has been used quite heavily to study vaccine responses to influenza A virus vaccines. The manipulability of the model has allowed description of the role for specific cell types in live-attenuated influenza virus vaccine responses [13], and differential roles for memory or naive B cell populations based on live-attenuated or inactivated vaccine formulation [14]. It has been demonstrated that adenoviral vector induced antibody responses can be measured, but no specific investigation of the regulation of these responses was performed [13]. Thus, this organoid model has promise for unravelling pathways regulating adenoviral vector induced humoral responses.

We utilized this human in vitro tonsil organoid model to investigate the processes that regulate the induction of T cell, B cell and antibody responses to the ChAdOx1 nCoV-19 vaccine. The tonsil organoid system demonstrated cellular, humoral and innate responses to adenovirus vaccines, presenting a viable model for mechanistic studies. We discovered that plasmacytoid dendritic cells modulate the humoral response to adenovirus vector vaccines, mainly through secretion of type 1 IFN. Type 1 IFN augments antibody production through modulating both innate and adaptive responses, both directly and indirectly through IL-6 induction. These two cytokines also worked in concert to regulate the phenotype and function of vaccine-activated CD4$^+$ T cells.

## Materials and methods

### Ethics statement

Ethics approval was obtained from the Oxford Radcliffe Biobank research tissue bank ethics (reference 19/SC/0173). Pre-pandemic tonsil samples were collected prior to 2019 under the GI Biobank ethics (reference: 16/YH/0247). Written informed consent was received from all tissue donors, or their respective legal guardians, as appropriate. All work was performed in compliance with the principles of the Declaration of Helsinki (2008) and in accordance with relevant ethical regulations.

### Tonsil tissue collection

Whole human tonsils were collected from January 2022 to October 2023 from adult or pediatric patients undergoing tonsillectomy for sleep apnea or any other indication for elective tonsillectomy. Patients receiving immunosuppressive treatment, with severe inflammatory or exudative lesions on the tonsils, or with underlying immunocompromising disease were excluded. Beyond reason for surgery, donors were fully anonymized and no other clinical or demographic information was collected.

### Tonsil tissue processing

Tonsil tissue was processed as previously described [13]. Whole tonsils were collected in phosphate buffer saline (PBS) after surgery, then decontaminated by immersion in antimicrobial media (PBS, 1% penicillin, 1% streptomycin, 0.01% Normocin (Invivogen)) for ≥1 hour at 4°C. Tonsils were dissected and ground through a 70-micron strainer to obtain single-cell suspensions. To decrease cell debris, cells were isolated by Ficoll density gradient separation and washed with complete medium (RPMI-1640 with glutamine, 10% fetal bovine serum (FBS), 1% nonessential amino acids, 0.01% Normocin, 1% insulin/transferrin/selenium). Cells were counted, frozen in FBS with 10% dimethyl sulfoxide (DMSO), and stored at ≤ −150°C.

Pre-pandemic tonsil samples were processed and stored as previously described [17].

## Tonsil organoid culture

Tonsil organoid cultures were performed as previously described [13]. Cell aliquots were thawed and resuspended in complete media at $4 \times 10^7$ cells/mL. Cells were plated in 96-well permeable (0.4-µm pore size) transwell polycarbonate membrane plates (Corning), with the upper chamber containing $1 \times 10^6$ or $2 \times 10^6$ cells, depending on the experiment, and the lower chamber containing media. Culture media was supplemented with 1 µg/mL recombinant human B cell-activating factor (BAFF; BioLegend). Cultures were stimulated using ChAdOx1 nCoV-19 (expressing the SARS-CoV-2 spike Wuhan variant), ChAdOx1 GFP, or human adenovirus 5 (Ad5) nCoV-19 at a multiplicity of infection (MOI) of 1000 viral particles/ cell. ChAdOx1 nCoV-19 was used either as discarded clinical product (OUH Pharmacy) or produced by the Pandemic Sciences Institute (University of Oxford) Viral Vector Core Facility. ChAdOx1 GFP and Ad5 nCoV-19 were produced by the Pandemic Sciences Institute Viral Vector Core Facility (University of Oxford). Cultures were incubated at 37°C and 5% $CO_2$. Culture media in the bottom chamber was replaced with fresh media every three days.

pDC-depletion was done using magnetic separation using the CD303 (BDCA-2) MicroBead Kit, per manufacturer's instructions (Miltenyi Biotec). Tonsil cells were incubated with biotinylated anti-BDCA-2 antibodies and subsequently with anti-biotin magnetic beads. Cells were then passed through LS separation columns (Miltenyi Biotec) placed on magnetic stands. pDC-depleted cells were collected from the flowthrough and cultured as described above. IFN-α and IL-6R blockade treatment used B18R (eBioscience, final concentration: 10 µg/mL) and anti-human IL-6R (InVivoSim, Bio X Cell, final concentration: 10 µg/mL). IFN-α supplementation used IFN-α (Sigma-Aldrich) with a final concentration of 50 ng/ml.

B cell separation was done using magnetic separation using the CD19 Microbead Kit, per manufacturer's instructions (Miltenyi Biotec). Tonsil cells were incubated with magnetic beads conjugated to anti-CD19 antibodies. Cells were then passed through LD separation columns (Miltenyi Biotec) placed on magnetic stands. B cell depleted cells were collected from the flowthrough, while selected B cells were collected by flushing the columns after removing LD columns from the magnetic stands. B cell depleted cells and selected B cells were then cultured as described above.

## Antibody measurement

Culture supernatants were harvested from the lower chamber of the transwell plates, supernatants were collected every three days starting from day 6 of culture alongside the media change. Specific IgG and IgM antibody levels were measured in culture supernatants by ELISA using LEGEND MAX SARS-CoV-2 Spike S1 Human IgG ELISA kit (BioLegend) and LEG-END MAX SARS-CoV-2 Spike S1 Human IgM ELISA kit (BioLegend), respectively, per manufacturer's instructions. Total IgG levels were measured using LEGEND MAX Total IgG ELISA kit (BioLegend) per manufacturer's instructions. Samples with calculated antibody titers below the concentration of the lowest standard were reported as half of the lowest standard.

## Generation of B cell tetramers

Antigen-specific B cells were identified using binding to fluorescently labelled SARS-CoV-2 spike S1 region tetramers. S1 tetramers were generated as previously described [18]. Tetramerization was done through addition of PE- or BV421-conjugated streptavidin (BioLegend) to biotinylated S1 proteins (BioLegend). Streptavidin was added incrementally to a final streptavidin:S1 molar ratio of 1:5, which was done in five additions with 20 minutes incubation at 4°C in between.

## Flow cytometry

Tonsil organoid cells were harvested from the upper chamber by washing with PBS. Cells were then washed twice in FACS buffer (PBS + 0.05% BSA + 2mM EDTA) before incubation with the surface staining cocktail (reagents listed in Table 1) for 30 minutes light-protected at 4°C. Cells were washed twice in FACS buffer, fixed using BD Cytofix (BD Biosciences) or BD Cytofix/Cytoperm (BD Biosciences) for 20 minutes at 4°C, and washed twice with FACS buffer. Samples were stored

at 4°C before running on the flow cytometer. Flow cytometry was done using BD LSRFortessa Cell Analyzer or BD LSR II Flow Cytometer (BD Biosciences).

The gating strategies used are shown in S1 Fig.

## T cell assays

For T cell assays, tonsil organoids were plated as described above and cultured for 7 days. Brefeldin A was added to cultures at 16–20 hours before harvest to enhance intracellular cytokine detection. Cells were harvested, washed, and incubated in the surface staining cocktail (reagents listed in Table 1) as described above. Cells were then fixed and permeabilized by incubation in BD Cytofix/Cytoperm (BD Biosciences) for 20 minutes at 4°C. Fixed cells were incubated with

**Table 1. List of flow cytometry reagents.**

| Fluorophore | Marker | Clone | Species origin | Dilution (1 in X) | Supplier |
|---|---|---|---|---|---|
| AF488 | SARS-CoV-2 spike | 3G8 | Mouse | 100 | Biotechne |
| PE | PD-L1 | MIH2 | Mouse | 50 | BioLegend |
| PE-CF594 | CD11c | B-ly6 | Mouse | 100 | BD Biosciences |
| PE-Cy7 | CD19 | M5E2 | Mouse | 100 | BioLegend |
| APC | CD16 | 3G8 | Mouse | 100 | BioLegend |
| AF700 | HLA-DR | L243 | Mouse | 100 | BioLegend |
| BV421 | CD80 | 2D10 | Mouse | 100 | BioLegend |
| BV510 | CD123 | 6H6 | Mouse | 100 | BioLegend |
| BV650 | CD14 | HIB19 | Mouse | 100 | BioLegend |
| BV711 | CD3 | OKT3 | Mouse | 100 | BioLegend |
| BV785 | CD45 | HI30 | Mouse | 100 | BioLegend |
| FITC | CD38 | HIT2 | Mouse | 100 | BioLegend |
| PE | S1-tetramer | -- | -- | 50 | BioLegend |
| PerCP-Cy5.5 | CD27 | O323 | Mouse | 100 | BioLegend |
| PE-Cy7 | IgM | MHM-88 | Mouse | 100 | BioLegend |
| AF700 | IgD | IA6–2 | Mouse | 100 | BioLegend |
| BV421 | S1-tetramer | -- | -- | 50 | BioLegend |
| BV650 | CD19 | HIB19 | Mouse | 100 | BioLegend |
| BV421 | IL-6 | MQ2-13A5 | Rat | 100 | BD Biosciences |
| FITC | CD3 | UCHT1 | Mouse | 100 | BioLegend |
| PE-CF594 | CCR7 | G043H7 | Mouse | 50 | BioLegend |
| AF700 | CD8 | SK1 | Mouse | 100 | BioLegend |
| APC-Cy7 | CD14 | M5E2 | Mouse | 100 | BioLegend |
| APC-Cy7 | CD19 | HIB19 | Mouse | 100 | BioLegend |
| BV510 | CD45RA | HI100 | Mouse | 100 | BioLegend |
| BV605 | CXCR5 | J252D4 | Mouse | 50 | BioLegend |
| BV711 | CD4 | OKT4 | Mouse | 100 | BioLegend |
| BV785 | PD-1 | EH12.2H7 | Mouse | 100 | BioLegend |
| PE | CD40L | 24-31 | Mouse | 100 | BioLegend |
| PE-Cy7 | IL-2 | MQ1-17H12 | Rat | 100 | BioLegend |
| APC | IFN-γ | B27 | Mouse | 100 | BioLegend |
| BV421 | CD69 | FN50 | Mouse | 100 | BioLegend |
| BV650 | TNF | MAb11 | Mouse | 100 | BioLegend |
| Near-infrared | Live/dead | -- | -- | 400 | Invitrogen |

the intracellular staining cocktail (reagents listed in Table 1) for 30 minutes at 4°C and washed twice before running the samples on the flow cytometer. Flow cytometry was done using BD LSRFortessa Cell Analyzer or BD LSR II Flow Cytometer (BD Biosciences).

### Cytokine measurements

Cytokine levels were measured from culture media supernatants harvested 24 hours after plating using LegendPlex Human Inflammation Panel 1 (BioLegend), per manufacturer's instructions. Briefly, supernatants were incubated with detection beads coated with antibodies against the panel cytokines for two hours. Beads were then incubated with biotinylated detection antibodies and subsequently with PE-conjugated streptavidin. Beads were then detected by flow cytometry (BD LSRFortessa Cell Analyzer), then converted back to cytokine concentrations using Qognit software (BioLegend).

### Data analysis

Analysis of flow cytometry data was performed using FlowJo version 10.8.1. Statistical analyses were performed in GraphPad Prism version 10. Comparison of different donor groups were done using a nonparametric Mann-Whitney U test. Paired samples (comparison of different treatments for matched/pairs donors) were done using Wilcoxon matched pairs signed rank test (for two groups) or using Friedman test with Dunn's multiple comparisons test (for three or more groups). Regression analysis was performed using simple linear regression.

## Results

### Adenovirus vector vaccines induce humoral responses in tonsil organoid cultures

Our first aim was to validate that the tonsil organoid culture system could robustly model humoral responses to adenovirus vector vaccines. To achieve this, we stimulated tonsil organoid cultures with ChAdOx1 nCoV-19 over 14 days of culture (experimental schematic in S2A Fig). Viability and cell recovery remained consistent over the culture period (S2B Fig).

To validate that the system could induce specific antibody production in response to vaccine stimulation, anti-SARS-CoV-2 S1 IgG was measured in the culture supernatants. Anti-S1 IgG production was very low (<10 ng/mL) or undetectable in unstimulated cultures from 18/22 post-pandemic donors (Fig 1A). ChAdOx1 nCoV-19 stimulation induced robust S1-specific antibody responses in the majority of post-pandemic donors (n = 13/16; Fig 1A). To test for non-specific IgG induction by the ChAdOx1 vector, ChAdOx1 GFP was used as a control. ChAdOx1 GFP did not induce anti-S1 IgG production in any of the donors tested (Fig 1A). Peculiarly, a small fraction of post-pandemic donors spontaneously produced anti-S1 IgG without stimulation and ChAdOx1 stimulation suppressed specific antibody production (S2C Fig). Given their unsuitability for addressing the research question, samples with spontaneous anti-S1 IgG at baseline were excluded from subsequent analyses. Stimulation of tonsils collected prior to the COVID-19 pandemic ("pre-pandemic") did not induce S1-specific IgG production (Fig 1B). This was not simply due to delayed kinetics as extending the culture to 27 days still did not induce anti-S1 IgG production in the pre-pandemic donors tested (S2D Fig).

We also measured total IgG production from the tonsil organoid supernatants, discovering that total IgG levels were much higher than anti-S1 IgG levels (S2E Fig). These findings are in line with a previous study reporting that tonsil cells from pediatric tonsillectomies produce IgG at comparable levels in 7-day cultures without stimulation [19]. Interestingly, ChAdOx1 nCoV-19 stimulation suppressed total IgG production (S2E Fig). One possible explanation is the increased metabolic requirements in activated B cells [20], resulting in resource competition and reduced non-specific IgG production. This phenomenon may also explain why ChAdOx1 nCoV-19 suppresses anti-S1 IgG antibody production in donors with spontaneous anti-S1 IgG production.

Although we could not observe class-switched antibody production from the pre-pandemic donors, we hypothesized that the pre-pandemic donors may produce specific IgM antibodies as an earlier product of naive B cell activation. In

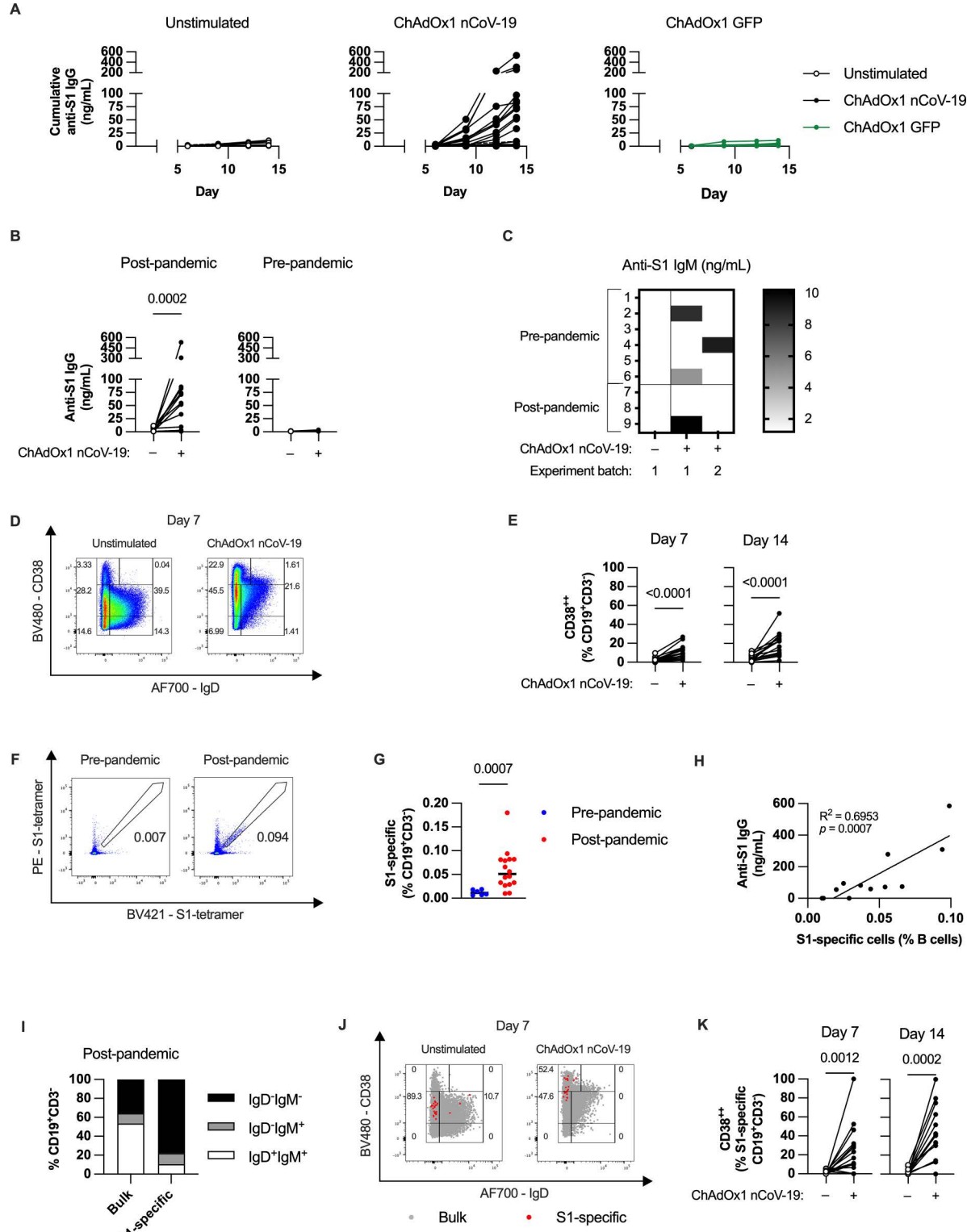

**Fig 1. ChAdOx1 nCoV-19 induces specific antibody production and B cell activation in post-pandemic tonsil organoids. (A)** Cumulative anti-S1 IgG production over time in unstimulated tonsil organoids (*left*) or in response to ChAdOx1 nCoV-19 (*middle*) or ChAdOx1 GFP (*right*). **(B)** Cumulative anti-S1 IgG production with ChAdOx1 nCoV-19 stimulation in tonsils collected post- and pre-COVID-19 pandemic. **(C)** Cumulative anti-S1 IgM

production on day 14 in pre- and post-pandemic tonsil organoids over repeated experiments. **(D)** Representative FACS plot of B cells from unstimulated and ChAdOx1 nCoV-19-stimulated organoids on day 7 of culture. **(E)** Total B cell activation measured by % of CD38++ cells in response to ChAdOx1 nCoV-19 in post-pandemic donors. **(F)** Representative FACS plot of S1-tetramer binding of B cells from pre- and post-pandemic donors. **(G)** Percentage of S1-specific B cells in pre- and post-pandemic donors. **(H)** Association of S1-tetramer binding B cell frequency on day 0 and cumulative anti-S1 IgG titers after 14 days. **(I)** Surface IgD and IgM expression on S1-specific B cells in post-pandemic donors. **(J)** Representative FACS plot of S1-specific B cells overlaid on total B cells on Day 7 of organoid culture from a post-pandemic donor. **(K)** S1-specific B cell activation measured by % of CD38++ cells in response to ChAdOx1 nCoV-19 in post-pandemic donors. Data in A-B are combined from 5 experiments with 24 donors, C from 2 experiments with 9 donors, E from 4 experiments with 16 donors, G from 5 experiments with 22 donors, H from 3 experiments with 12 donors, I and K from 4 experiments with 16 donors. Each symbol represents an individual donor. Values in B, E and K were compared using Wilcoxon matched pairs signed rank test, values in G were compared using Mann-Whitney test. Analysis in H was simple linear regression analysis.

contrast to IgG production, anti-S1 IgM was produced in both post- and pre-pandemic donors in a variable manner across two experiment batches (Fig 1C). The high variability between technical replicates suggests that specific IgM production in the model is stochastic, possibly depending on the presence in the culture well of S1-specific naïve B cells, which are present at low frequencies [21].

After validating antibody production upon stimulation, we measured B cell activation using flow cytometry (S1A Fig). We compared B cell subsets based on IgD and CD38 expression as naïve and activation markers, respectively [22]. ChAdOx1 nCoV-19 decreased the percentage of IgD+ B cells and increased CD38 expression on B cells (Fig 1D). Overall, the most striking change was an increase in IgD-CD38++ B cells, a population commonly identified as antibody secreting cells (ASCs) [22], which encompasses both plasmablasts and plasma cells (Fig 1D and 1E). A similar increase in this population was observed for ChAdOx1 GFP (S2F Fig), but did not induce S1-specific antibody production (Fig 1A). Thus, ChAdOx1 vectors appear to induce a proportion of memory B cells (IgD-CD38+) to convert to the ASC phenotype.

The IgD-CD38++ population had higher CD27 expression levels compared to other B cell subsets within the same condition (S2G Fig), in line with previous observations for ASCs [22]. Activated memory B cells upregulate CD27 expression during plasmablast differentiation [23], converting from a CD27+ to CD27++ phenotype, and thus is often used as a marker of ASCs. Upon stimulation, there was lower average CD27 expression compared to the pre-existing IgD-CD38++ subset in unstimulated organoids (S2G Fig), suggesting the conversion of recently activated memory B cells that have yet to convert to CD27++.

Next, we identified antigen-specific B cells using fluorescently-labelled S1 tetramers. Post-pandemic donors showed a clear population of S1-specific B cells, while pre-pandemic donors did not (Fig 1F and 1G). The cumulative anti-S1 IgG production over 14 days associated positively with pre-existing frequency of S1-specific B cells ($R^2 = 0.6953$, p = 0.0007, Fig 1H). The non-responding post-pandemic donors had lower percentages of S1-specific B cells, likely explaining the undetectable specific antibody production. In post-pandemic donors, the S1-specific B cells were mainly composed of switched memory (IgD-IgM-) B cells compared to the total B cell population (Fig 1I). The percentage of detectable S1-specific B cells decreased with ChAdOx1 nCoV-19 stimulation (S2H Fig), possibly due to activated cells downregulating their B cell receptors (BCR) as they differentiate into ASCs. However, S1-specific cells were detectable, and were highly activated in response to stimulation (Fig 1J and 1K). These findings suggest that S1-specific IgG production, but not IgM production, in response to stimulation is associated with detectable S1-specific memory B cells. Most importantly, the combination of these data gave us high confidence in the utility of the model for probing pathways regulating ChAdOx1-induced humoral immunity.

### Plasmacytoid dendritic cells (pDCs) contribute to the early response to ChAdOx1 nCoV-19 through antigen expression and cytokine secretion

Next, we investigated the early innate response to ChAdOx1 nCoV-19 in the tonsil organoid model after 24 hours in culture. As S1-specific IgG production was only induced in organoids from post-pandemic donors, only these donors were

used for mechanistic studies. We focused on identifying the cells that are transduced and activated by ChAdOx1 by flow cytometry (S1B Fig). To identify cells transduced by the vector, we measured cells expressing either the S1 spike peptide (for ChAdOx1 nCoV-19) or GFP (for ChAdOx1 GFP) by flow cytometry. No significant changes in the tonsil cell compositions were observed with freeze-thawing (S3A Fig). Following exposure, all cells showed an increase in transgene-expressing cells, with varying frequencies (Figs 2A and S3B). The highest transduction rates were seen for dendritic cells (DCs), especially plasmacytoid dendritic cells (pDCs) (Fig 2A). The number of pDCs decreased with ChAdOx1 nCoV-19

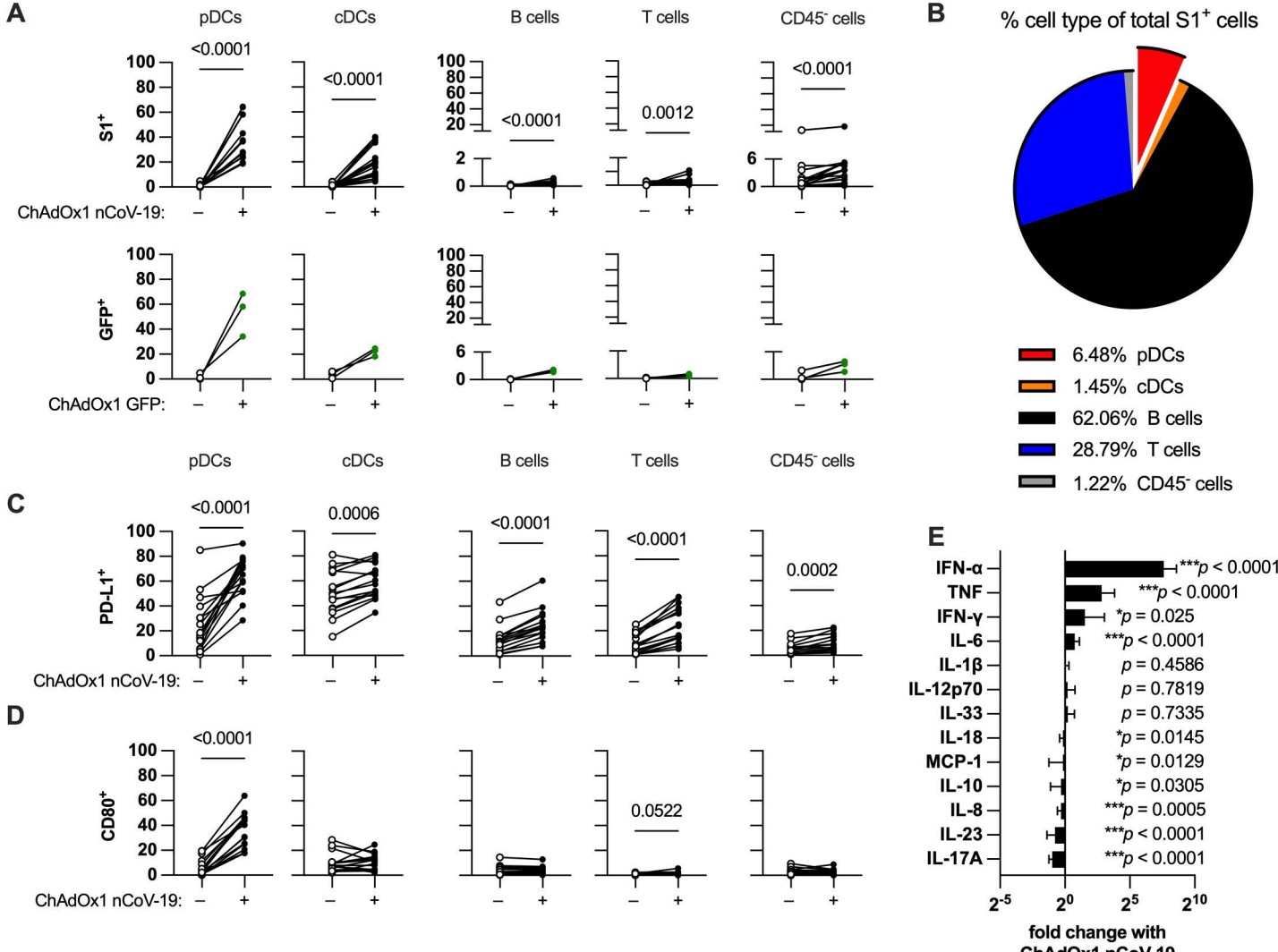

**Fig 2. pDCs contribute to the early response to ChAdOx1 nCoV-19 through antigen expression and cytokine secretion. (A)** Transduction rates of different cell types in the tonsil organoid culture system. Transduction was measured based on detection of ChAdOx1 transgene (either S1 spike or GFP) by flow cytometry. **(B)** Average fraction of each cell type as a proportion of all transduced cells. **(C-D)** Innate immune cell activation by ChAdOx1 nCoV-19 as measured by percentage of cells expressing surface PD-L1 **(C)** or CD80 **(D)**. **(E)** Relative change in cytokine levels in tonsil organoid media supernatants with ChAdOx1 nCoV-19 stimulation compared to unstimulated (each bar shows mean±SD). All results were from cell/media harvest at 24 hours after plating. Data in A-E are combined from 4 experiments with a total of 16 donors. Each symbol represents an individual donor. Values in A and C-D were compared using Wilcoxon matched pairs signed rank test. Values in E were compared using Wilcoxon matched pairs signed rank test from unstimulated compared to ChAdOx1 nCoV-19-stimulated wells (data in S3D Fig).

stimulation, likely due to death following transduction (S3C Fig). Both ChAdOx1 nCoV-19 and ChAdOx1 GFP demonstrated similar patterns of transduction, with higher detection rates for GFP, likely due to ease of detecting GFP expression. Monocytes were highly transduced (66–100%) but were not included in subsequent analyses as their scarcity in tonsil made quantification unreliable (0–5 cells/well). Conventional DCs (cDCs) and CD45⁻ cells (presumably fibroblasts) were also transduced with some efficiency (mean % positive for each group). This pattern of transduction is highly consistent with what has been reported in peripheral blood mononuclear cells (PBMCs) [24].

Despite having the highest transduction efficiency, pDCs only contributed a small fraction of total transduced cells (6.5%), due to the relatively small size of the pDC population in the culture (~1% of all cells) (Fig 2B). By contrast, B cells were the main contributor of transduced cells despite very low transduction rates due to the high frequency of B cells in the culture system (~65% of all cells).

We next evaluated immune activation through the expression of PD-L1 and CD80. ChAdOx1 nCoV-19 stimulation upregulated PD-L1 expression on all cell populations, while CD80 was only induced on pDCs (Fig 2C and 2D). Other than pDCs, most immune cell types had higher PD-L1 expression rates than their transduction rates, suggesting indirect activation of non-transduced cells, likely through cytokines.

We measured ChAdOx1 nCoV-19 induced cytokine secretion at 24 hours. ChAdOx1 nCoV-19 induced significant production of several pro-inflammatory cytokines, including IFN-α, TNF, IFN-γ, and IL-6, with the greatest induction of IFN-α (Figs 2E and S3D). The induced cytokines are consistent with previous studies on patient samples and PBMC stimulation using ChAdOx1 nCoV-19 [25].

## pDCs modulate the humoral response to ChAdOx1 nCoV-19, primarily through type 1 IFN

Given the high transduction and activation of pDCs as well as the IFN-α induced, we hypothesized that pDCs were an important early immune modulator of response to adenovirus vector vaccines. Thus, we performed the organoid culture with pDC depletion (see methods). The depletion process removed 98.6-99.9% of pDCs, while the percentage of other cells remained stable except for a slight increase of CD45⁻ cells (S4A Fig).

pDC-depletion dramatically impaired S1-specific IgG induction, confirming the modulatory function of pDCs on antibody production (Fig 3A). pDC-depletion also suppressed total IgG production (S4B Fig). In accordance with the decreased antibody production, both total and S1-specific B cell activation was also dampened by pDC-depletion (Fig 3B and 3C). pDC-depletion did not significantly affect the overall antigen availability as shown by the total frequency of transduced cells in the culture (Fig 3D) nor did it alter transduction rates for other cell types (Fig 3E). However, early activation of other immune cells (cDCs, B cells, T cells), as assessed by PD-L1 expression, decreased with pDC-depletion (Fig 3F). There was minimal impact on CD80 expression, as no cells showed clear induction of CD80 (S4C Fig). As expected, pDC-depletion dramatically reduced levels of IFN-α (Figs 3G and S4D). However, it also resulted in a reduction of the other induced pro-inflammatory cytokines TNF, IFN-γ, and IL-6. These results demonstrate that pDCs are critical modulators of the humoral responses to adenovirus vectors.

## IFN-α is the main mediator of pDC-mediated modulation of the humoral response

To investigate whether the effect of pDCs on the humoral response was mediated by type 1 IFN, as a source of antigen, or both, we investigated the effect of blocking type 1 IFN on the culture. Type 1 IFN blockade was sufficient to phenocopy pDC-depletion, significantly reducing S1-specific IgG secretion (Fig 4A), total B cell activation (Fig 4B), and S1-specific B cell activation (Fig 4C) in all donors.

With regards to innate responses, blocking type 1 IFN increased the overall number of transduced cells (Fig 4D). The increase appears to be mediated by a particular increase in pDC transduction rates and a modest increase for B cells (S5A Fig). Interestingly, despite increased transduction rates, PD-L1 expression on most cell types decreased with type 1 IFN blockade (Fig 4E). This suggests that upregulation of surface PD-L1 expression on pDCs is mediated by autocrine

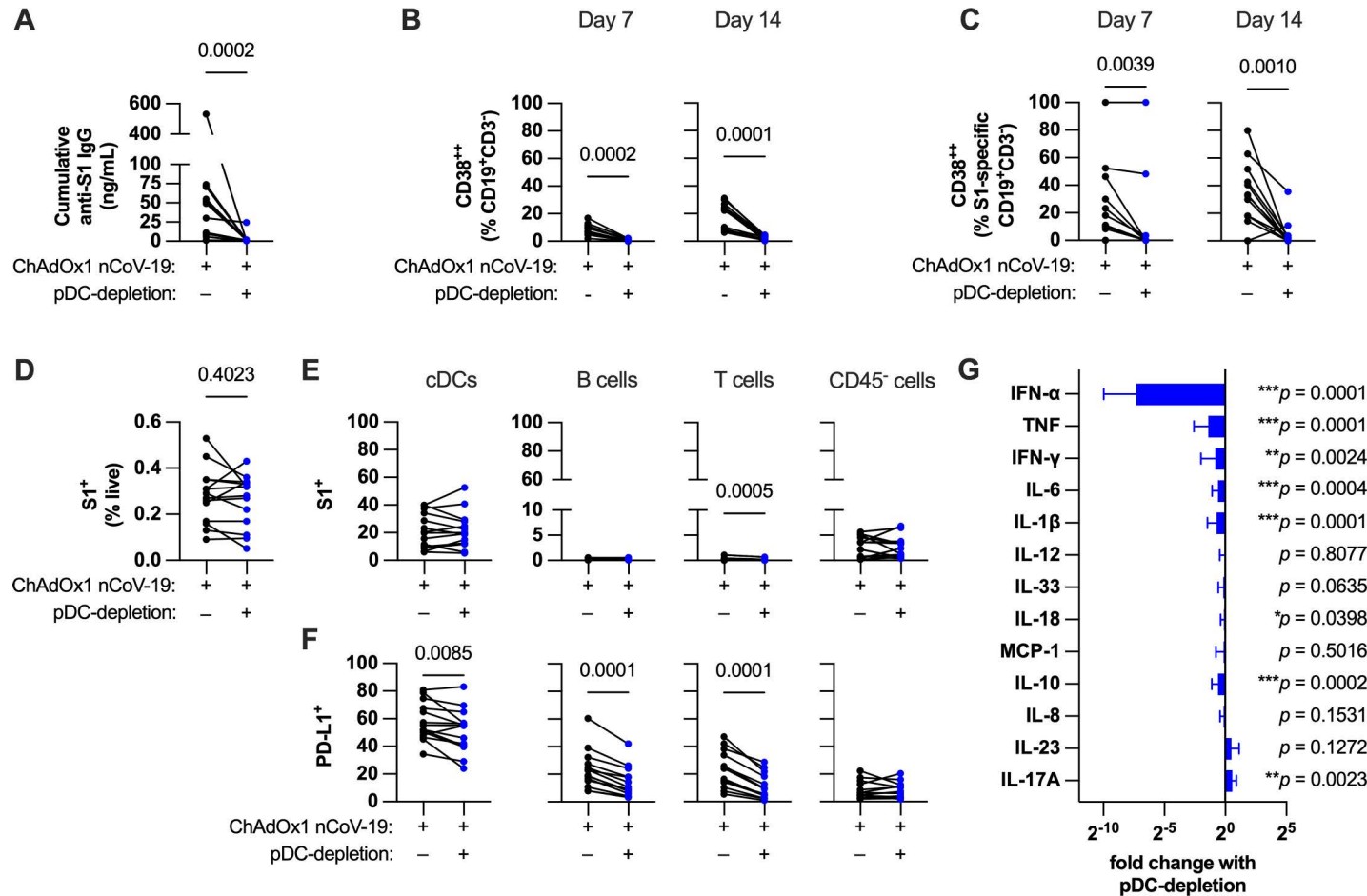

**Fig 3. pDC-depletion impairs both humoral and innate responses to ChAdOx1 nCoV-19. (A)** Cumulative anti-S1 IgG production in ChAdOx1 nCoV-19-stimulated post-pandemic tonsil organoids with or without pDC-depletion after 14-day culture. **(B-C)** Total **(B)** and S1-specific **(C)** B cell activation in ChAdOx1 nCoV-19-stimulated tonsil organoids with or without pDC-depletion. **(D)** Percentage of total transduced cells after 24-hour culture in ChAdOx1 nCoV-19-stimulated tonsil organoids with or without pDC-depletion. **(E)** Transduction rates of different cell types in ChAdOx1 nCoV-19-stimulated tonsil organoids with or without pDC-depletion. **(F)** Innate activation of different cell types based on surface PD-L1 in ChAdOx1 nCoV-19-stimulated tonsil organoids with or without pDC-depletion. **(G)** Relative change in cytokine levels in ChAdOx1 nCoV-19-stimulated tonsil organoid supernatants with or without pDC-depletion (each bar shows mean±SD). Data in A-G are combined from 4 experiments with a total of 14 donors. Each symbol represents an individual donor. Values in A-F were compared using Wilcoxon matched pairs signed rank test. Values in G were compared using Wilcoxon matched pairs signed rank test from ChAdOx1 nCoV-19-stimulated untreated wells compared to stimulated wells with pDC-depletion (data in S4D Fig).

type 1 IFN and that transduction alone is insufficient to activate the pDCs. Induction of TNF, IFN-γ and IL-6 were marginally but consistently decreased with type 1 IFN blockade (S5B Fig), again similar to pDC depletion.

While these data clearly supported a model where pDC-derived IFN-α is important for ChAdOx1 nCoV-19-induced humoral responses, it was unclear if pDC-derived antigen was also important. To investigate this, we determined if addition of exogenous IFN-α was sufficient to rescue pDC-depleted cultures. Adding exogenous recombinant IFN-α to pDC-depleted cultures restored S1-specific IgG antibody secretion (Fig 4F) and B cell activation (Fig 4G and 4H). Adding IFN-α without ChAdOx1 nCoV-19 stimulation caused B cell activation (CD38++) but no anti-S1 IgG secretion (S5C Fig), further confirming that cognate antigen is required for specific antibody secretion. IFN-α supplementation did not affect overall percentage of transduced cells (S5D Fig).

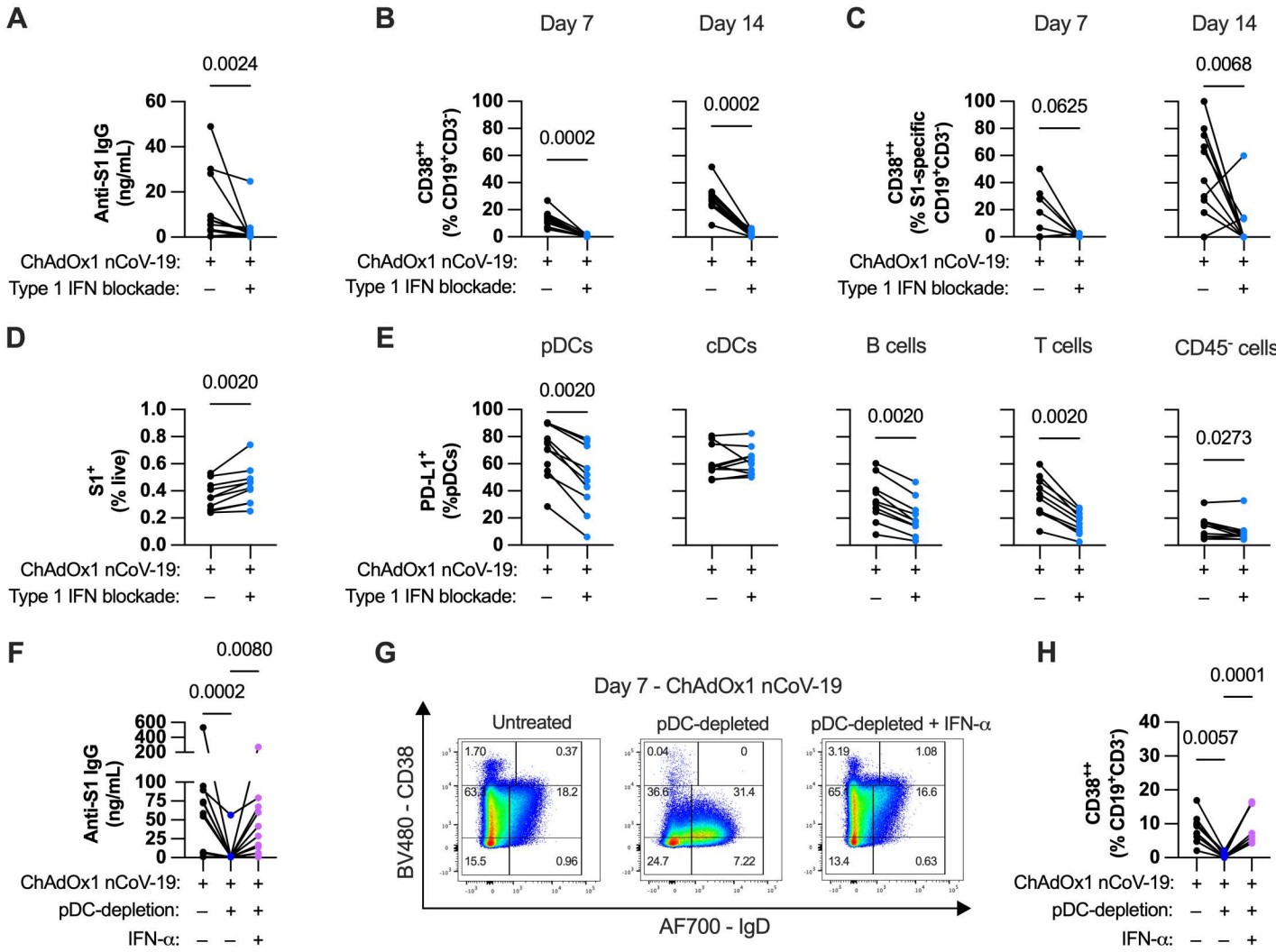

**Fig 4. pDCs modulate the humoral response to adenoviral vectors primarily through type 1 IFN. (A)** Cumulative anti-S1 IgG production in ChAdOx1 nCoV-19-stimulated tonsil organoids with or without type 1 IFN blockade after 14-day culture. **(B-C)** Total **(B)** and S1-specific **(C)** B cell activation levels in ChAdOx1 nCoV-19-stimulated tonsil organoids with or without type 1 IFN blockade. **(D)** Percentage of total transduced cells after 24-hour culture in ChAdOx1 nCoV-19-stimulated tonsil organoids with or without type 1 IFN blockade. **(E)** Innate activation of different cell types based on surface PD-L1 in ChAdOx1 nCoV-19-stimulated tonsils organoids with or without type 1 IFN blockade. **(F)** Cumulative anti-S1 IgG production in ChAdOx1 nCoV-19-stimulated tonsil organoids with pDC-depletion and with or without IFN-α supplementation. **(G)** Representative FACS plot of total B cells in pDC-depleted ChAdOx1 nCoV-19-stimulated tonsil organoid cultures and with or without IFN-α supplementation. **(H)** Total B cell activation levels in pDC-depleted ChAdOx1 nCoV-19-stimulated tonsil organoids and with or without IFN-α supplementation. Data in A-C are combined from 3 experiments with a total of 13 donors, D-E from 3 experiments with 10 donors, F and H from 3 experiments with 11 donors. Each symbol represents an individual donor. Values in A-E were compared using Wilcoxon matched pairs signed rank test, values in F and H using Friedman test with Dunn's multiple comparisons test.

For a comparison with a different adenovirus vector, we tested the innate and humoral responses in the tonsil organoid system using human adenovirus 5 (Ad5) encoding the same SARS-CoV-2 spike antigen (Ad5 nCoV-19). Ad5 nCoV-19 stimulation did not induce type 1 IFN nor IL-6 in the tonsil organoid system (S5E Fig), similar to findings from stimulation of human PBMCs or serum from vaccinated NHPs [24,26]. Ad5 nCoV-19 stimulation induced minimal production of spike-specific IgG (S5F Fig). However, type 1 IFN supplementation augmented Ad5-induced antibody production (S5F

Fig). Thus, type I IFN is a major regulator of Ad vector-induced antibody production, and its differential induction accounts for the differential antibody production by distinct vector serotypes in this model.

While cognate antigen is critical for ChAdOx1-stimulation of humoral responses (Fig 1A), these results suggest that pDCs do not modulate these responses by producing cognate antigen. Instead, they promote humoral responses primarily by producing IFN-α.

**pDC-derived type 1 IFN augments the humoral response to ChAdOx1 nCoV-19 partially by stimulating IL-6 production**

Depletion of pDCs reduced IL-6 levels in the culture (Fig 2G), and type 1 IFN supplementation of pDC-depleted cultures also restored the induction of IL-6 (Fig 5A), showing that type 1 IFN regulates IL-6 production. As IL-6 is a known modulator of B cell responses [27,28], we sought to investigate if pDCs modulated ChAdOx1-induced humoral immunity through IL-6 as an intermediary signal. First, we tested whether IL-6 regulated humoral responses to ChAdOx1. Blockade of IL-6 signaling using anti-IL-6R antibodies significantly, but partially, decreased S1-specific IgG production across all donors (Fig 5B). There was a trend of decreased B cell activation as well, although this did not reach statistical significance (Fig 5C and 5D).

As monocytes are one of the best described sources of IL-6 [29], but are essentially absent in tonsils (S2A Fig), we next sought to determine the cellular source of IL-6. Thus, we performed intracellular staining of IL-6 at 24 hours after ChAdOx1 nCoV-19 stimulation (Fig 5E). Most cell types produced IL-6 to varying levels even in the absence of stimulation, while cDCs had a trend towards increased IL-6 production following stimulation (Fig 5E and 5F), suggesting a distinct role for this population in vaccine-induced activation. Despite this, due to the low number of cDCs present in the culture, cDCs form less than 1% of total IL-6+ cells (Fig 5G). Although the percentage of IL-6-producing B cells is low as a proportion of total B cells, given the high frequency of B cells in the culture, B cells comprise most of the IL-6-producing cells (Fig 5G).

Consistent with the flow cytometry analysis, isolated B cells produced IL-6 in response to ChAdOx1 nCoV-19 stimulation (Fig 5H). However, B cell-depleted cultures also produced substantial IL-6, suggesting that non-B cell populations are also a relevant source (Fig 5H). Interestingly, despite both cultures producing IL-6, there were major differences in the level of IFN-α in the cultures (Fig 5I). B cell-depleted cultures had high levels of IFN-α production (consistent with the presence of pDCs) and there was a strong association between IFN-α levels and IL-6 levels (Fig 5J). By contrast, despite producing substantial IL-6, the purified B cell culture had negligible induction of IFN-α (<10 pg/mL) (Fig 5I), and thus IL-6 and IFN-α levels were unsurprisingly not related (Fig 5J). These findings suggest that both B cells and cDCs contribute to IL-6 induction by ChAdOx1 nCoV-19, but through type 1 IFN-independent and -dependent mechanisms, respectively.

**Type 1 IFN and IL-6 augment the cellular response to ChAdOx1 nCoV-19, including for follicular helper T (T$_{FH}$) cells**

Another important factor of vaccine immunity is the CD4+ T cell response. Thus, we sought to examine T cell activation (S1C Fig). In the tonsil organoid model, ChAdOx1 nCoV-19 induced CD4+ T cell activation (CD40L+CD69+) (Fig 6A and 6B). ChAdOx1 nCoV-19 induced production of IFN-γ, TNF, and IL-2, as measured by intracellular cytokine staining (Figs 6C and S6A). Similar T cell activation was observed in response to ChAdOx1 nCoV-19 or ChAdOx1 GFP stimulation (S6B and S6C Fig), suggesting simultaneous measurement of vector- and transgene-specific T cells. Thus, we could assess the activation and effector function of CD4+ T cells activated by ChAdOx1 nCoV-19.

Given that type I IFN and IL-6 modulated humoral responses, we sought to determine if these pathways also modulated the CD4+ T cell response. Although blocking type 1 IFN or IL-6 signaling did not affect the frequency of CD69+CD40L+ CD4+ T cells (Fig 6D), both treatments decreased cytokine production by CD4+ T cells in response to ChAdOx1 nCoV-19 (Fig 6E). This suggests that type 1 IFN and IL-6 both play a role in augmenting the CD4+ T cell effector response stimulated by ChAdOx1 nCoV-19.

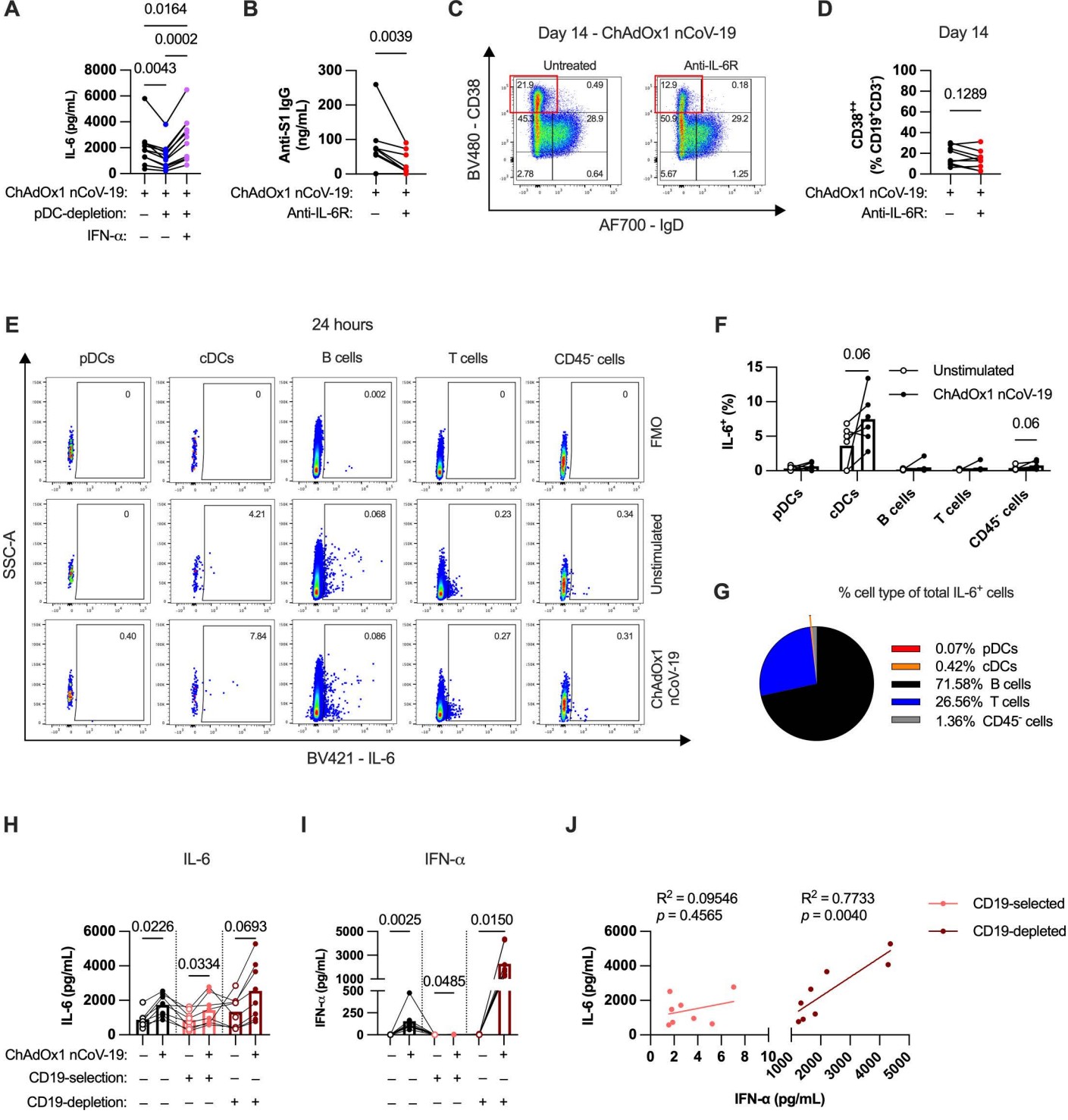

**Fig 5. pDC-derived type 1 IFN augments the humoral response to ChAdOx1 nCoV-19 partially through driving IL-6 production. (A)** IL-6 levels in ChAdOx1 nCoV-19-stimulated tonsil organoid supernatants with or without pDC-depletion and IFN-α supplementation. **(B)** Cumulative anti-S1 IgG production in ChAdOx1 nCoV-19-stimulated tonsil organoids with or without IL-6R blockade after 14-day culture. **(C-D)** Representative FACS plot **(C)** and group summary **(D)** of total B cells in tonsil organoid cultures with or without IL-6R blockade on day 14 of culture. **(E-F)** Representative FACS plots **(E)** and group summary **(F)** of IL-6 expression for each cell type in unstimulated and ChAdOx1 nCoV-19 stimulated organoids. **(G)** Average fraction of each cell type as a proportion of total IL-6-expressing cells. **(H-I)** IL-6 **(H)** and IFN-α **(I)** levels in ChAdOx1 nCoV-19-stimulated tonsil organoid cultures with or

without CD19-selection or CD19-depletion after 24 hours in culture. **(J)** Association between IL-6 and IFN-α levels in CD19-selected or CD19-depleted cultures after 24 hours in cultures with ChAdOx1 nCoV-19. Data in A are combined from 3 experiments with a total of 11 donors, B and D from 2 experiments with 9 donors, F-G from 2 experiments with a total of 6 donors, H-J from 2 experiments with a total of 8 donors. Each symbol represents an individual donor. Values in A were compared using Friedman test with Dunn's multiple comparisons test, values in B and D using Wilcoxon matched pairs signed rank test, values in F were compared using Wilcoxon matched pairs signed rank test. Values in H and I were compared using Friedman test with Dunn's multiple comparisons test, with comparisons between unstimulated and stimulated groups separately for each treatment. Regression analysis in J were done using simple linear regression.

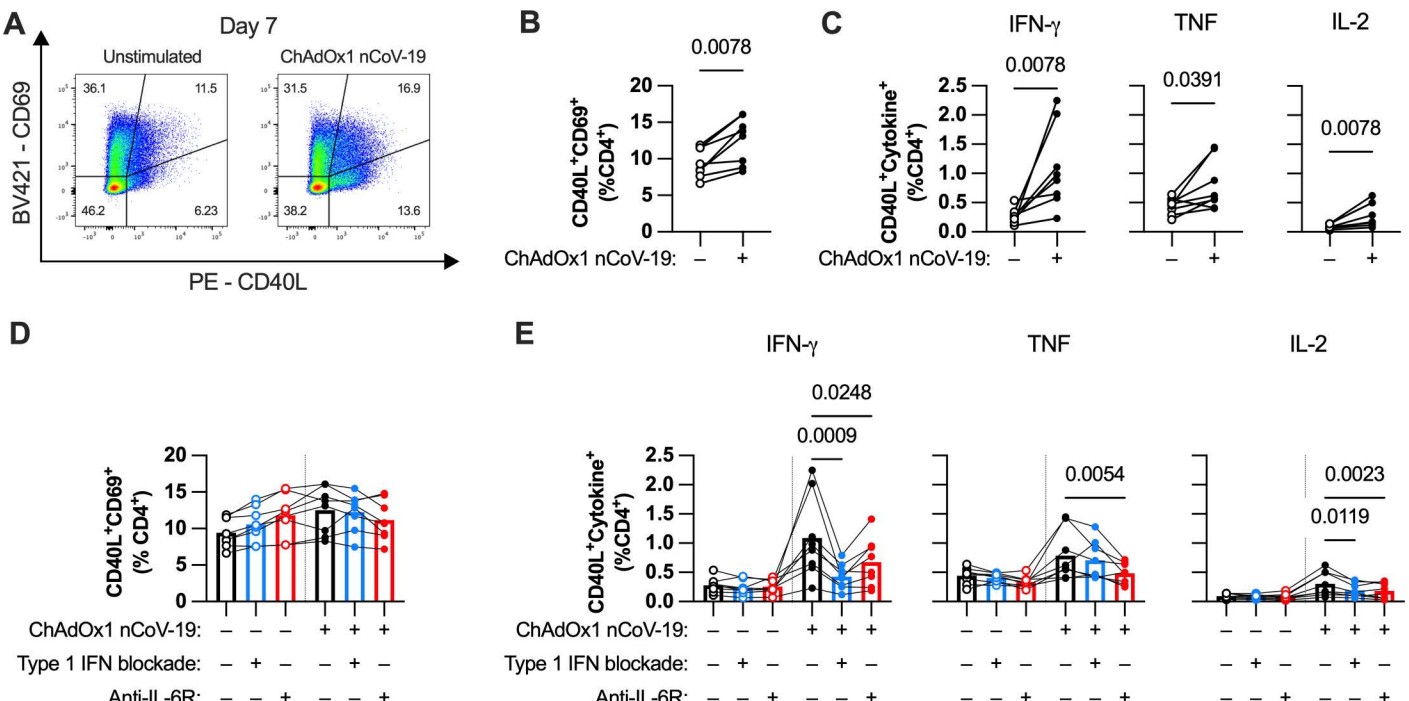

**Fig 6. Type 1 IFN and IL-6 augment the CD4⁺ T cell response to ChAdOx1 nCoV-19. (A-B)** Representative FACS plots of CD69 and CD40L expression on CD4⁺ T cells **(A)** and group summary **(B)** in ChAdOx1 nCoV-19-stimulated tonsil organoids. **(C)** Percentage of CD4⁺ T cells co-expressing CD40L and cytokines (IFN-γ, TNF, and IL-2) in ChAdOx1 nCoV-19-stimulated tonsil organoids. **(D)** Percentage of CD40L⁺CD69⁺ cells of CD4⁺ T cells in ChAdOx1 nCoV-19-stimulated tonsil organoids with or without type 1 IFN or IL-6R blockade. **(E)** Percentage of CD4⁺ T cells co-expressing CD40L and cytokines (IFN-γ, TNF, and IL-2) in ChAdOx1 nCoV-19-stimulated tonsil organoids with or without type 1 IFN or IL-6R blockade. All results were from cell harvest at 7 days after plating. Data are combined from 2 experiments with a total of 8 donors. Each symbol represents an individual donor. Values in B and C were compared using Wilcoxon matched pairs signed rank test. Values in D and E using Friedman test with Dunn's multiple comparisons test, with comparisons between stimulated groups with different treatments.

One particularly important CD4⁺ T cell subset in promoting B cell responses are follicular helper T (T$_{FH}$) cells, which regulate germinal center reactions [30]. We identified these cells by gating for PD-1⁺CXCR5⁺CD4⁺ T cells (Fig 7A). IL-6R blockade reduced the percentage of T$_{FH}$ cells both in unstimulated and stimulated wells (Fig 7A and 7B). This finding is likely due to the requirement of IL-6 to maintain the T$_{FH}$ phenotype [31]. However, with regards to function, blocking IL-6 signaling only resulted in a modest decrease in TNF-producing T$_{FH}$ and even a modest increase of CD40L⁺CD69⁺ T$_{FH}$ cells, suggesting less of an effect on T$_{FH}$ activation (Figs 7C, 7D and S6D). Conversely, blocking type 1 IFN did not affect the number of T$_{FH}$ cells, but had a strong inhibitory effect on both IFN-γ and TNF production by T$_{FH}$ cells. When examining activated T$_{FH}$ as a percentage of total CD4⁺ T cells, both treatments resulted in decreased frequencies of cytokine-producing T$_{FH}$ cells in the culture and no change in the percentage of CD40L⁺CD69⁺ T$_{FH}$ cells (Fig 7E and 7F). Overall,

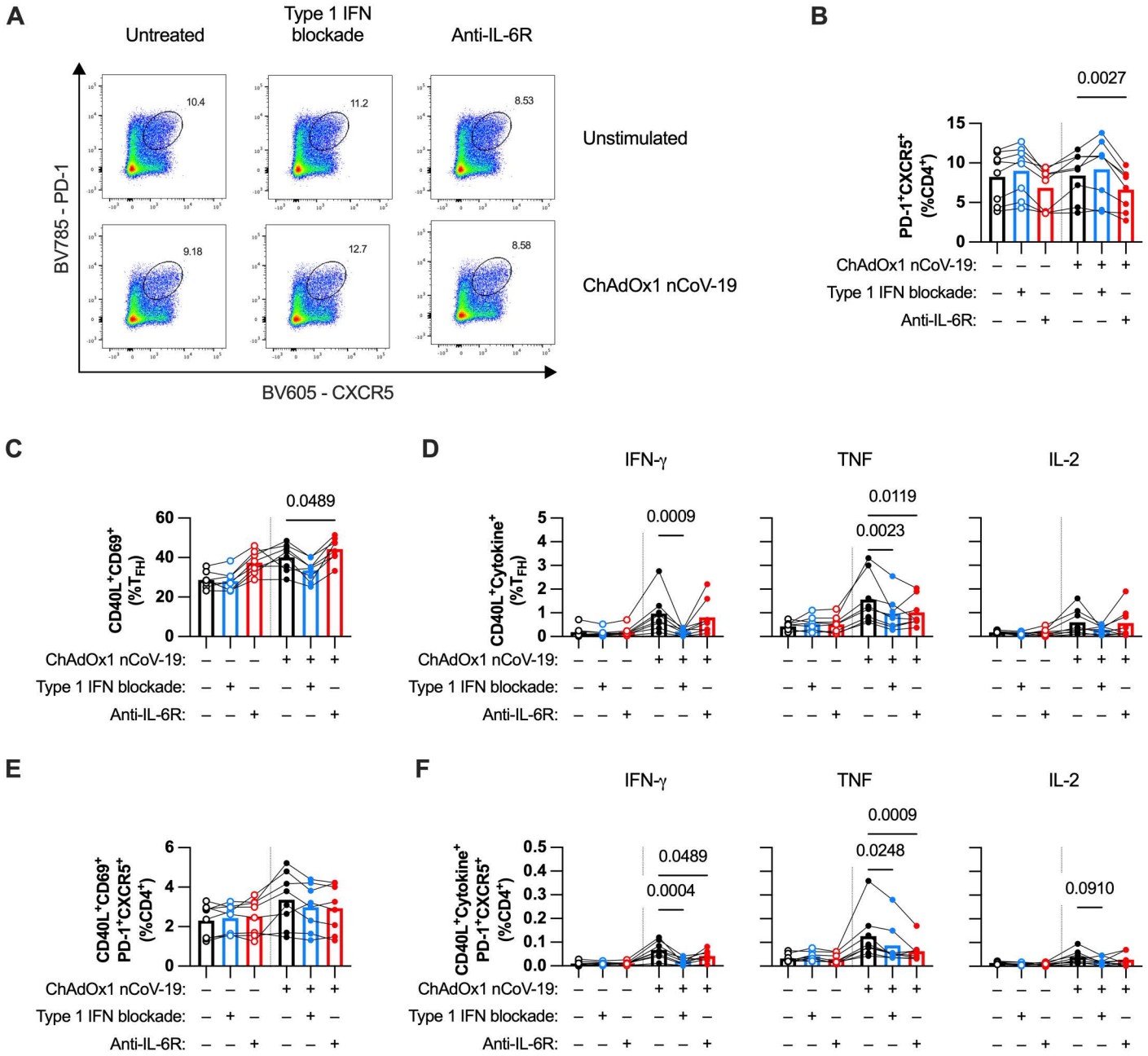

**Fig 7. Type 1 IFN and IL-6 augment T follicular helper (T_FH) cell responses to ChAdOx1 nCoV-19 through different mechanisms. (A)** Representative FACS plots of T_FH (PD-1+CXCR5+) gating on CD4+ T cells in ChAdOx1 nCoV-19-stimulated tonsil organoids with or without type 1 IFN or IL-6R blockade. **(B)** Percentage of T_FH cells (of total CD4+ T cells) with or without type 1 IFN or IL-6R blockade. **(C)** Percentage of CD40L+CD69+ cells as a fraction of CD4+ T_FH cells in ChAdOx1 nCoV-19-stimulated tonsil organoids with or without type 1 IFN or IL-6R blockade. **(D)** Percentage of cells expressing both CD40L and cytokines (IFN-γ, TNF, and IL-2) as a fraction of CD4+ T_FH cells in ChAdOx1 nCoV-19-stimulated tonsil organoids with or without type 1 IFN or IL-6R blockade. **(E)** Percentage of CD40L+CD69+ CD4+ T_FH cells as a fraction of total CD4+ T cells in ChAdOx1 nCoV-19-stimulated tonsil organoids with or without type 1 IFN or IL-6R blockade. **(F)** Percentage of CD4+ T_FH cells expressing both CD40L and cytokines (IFN-γ, TNF, and IL-2) as a fraction of total CD4+ T cells in ChAdOx1 nCoV-19-stimulated tonsil organoids with or without type 1 IFN or IL-6R blockade. All results were from cell harvest at 7 days after plating. Data combined from 2 experiments with a total of 8 donors. Each symbol represents an individual donor. Values in B-F were compared using Friedman test with Dunn's multiple comparisons test, with comparisons between stimulated groups with different treatments.

these data show that IL-6 and type 1 IFN both modulate T$_{FH}$ activation through different mechanisms: through maintaining T$_{FH}$ phenotype and augmenting cytokine production, respectively.

## Discussion

In this study we have validated the use of an in vitro human tonsil organoid system for studying the humoral responses to adenovirus vector vaccines. We could study vaccine-specific antibody production and B and T cell activation, as well as total B cell activation, which allows the investigation of the mechanisms underlying these responses. Our findings demonstrate the central role of plasmacytoid dendritic cells (pDCs) in modulating Ad vector humoral responses, primarily through type 1 IFN and indirectly through driving IL-6 signaling. Type 1 IFN and IL-6 also modulate CD4$^+$ T cell responses, including T$_{FH}$ cell responses, providing other potential mechanisms that regulate humoral responses to adenoviral vaccines. We summarize these findings in our model (Fig 8).

The in vitro tonsil organoid model has mainly been previously utilized for investigating responses to influenza vaccines [14–16]. Although previous studies focused on influenza vaccines, it has been shown that adenoviral vectors can induce antibody responses in the model [13]. We investigated further the cell-cell interactions and signaling modulating this humoral response and extended these investigations into the regulation of CD4$^+$ T cell, and specifically T$_{FH}$ cell, responses. Through using the system, we could model adaptive immune responses to ChAdOx1 nCoV-19 within lymphoid tissue and probe into mechanisms regulating these responses. Through combining cell depletions and cytokine blockade/supplementation, we could investigate intricate signaling interplay between different cells and cytokines, which revealed a critical pDC–IFN-α–IL-6 signaling axis. We believe this robustly validates the use of this model for studying entirely new classes of vaccines.

An important consideration with experimental models is identifying the aspects of the actual process that it captures. Our culture system appears to efficiently model a recall/boost response to adenoviral vectors rather than a "prime" response given that reliable detection of specific IgG antibody production requires pre-existing S1-specific B cells. However, IgM production could be measured but was stochastic likely due to limited frequency of naive B cells in the culture well. As such, our system appears to more robustly model a booster vaccination response in a previously exposed individual rather than a priming vaccination. The initial paper published on the model demonstrated priming of naive B

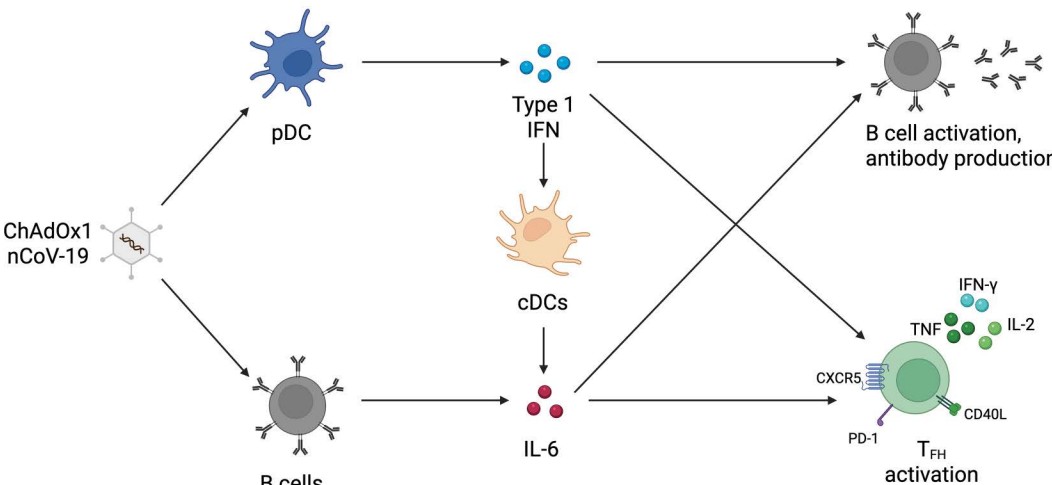

**Fig 8. Model of how ChAdOx1 nCoV-19 induces humoral and cellular responses.** Fig 8 was generated in BioRender (https://BioRender.com/xobun7x).

cell responses either by using larger culture wells or specifically enriching for naive B cells, increasing the overall number of naïve B cells in the system as well as the chances of having antigen-specific naïve B cells in the culture [13]. These parameters should be considered when designing and implementing this organoid model. As priming of naive B cells is a complex, multi-step process [32], other parameters may require optimization for this to be efficiently modeled in the culture system.

With our system, we discovered that pDCs are central modulators of humoral responses to ChAdOx1 nCoV-19. The role of pDCs has rarely been discussed in the context of humoral responses to adenovirus vector responses. A clinical study reported that the number of pDCs in blood are negatively correlated with antibody responses to ChAdOx1 nCoV-19 [33]. Elevated pDC frequency in the blood might indicate lower recruitment into lymph nodes after vaccination, supporting the role of pDCs in adenovirus vector humoral responses generated in SLOs. A previous study using the tonsil organoid model also reported a sharp decrease in humoral responses towards LAIV with pDC-depletion [13], suggesting that pDCs may also play a role in responses to multiple vaccine platforms.

Our findings show type 1 IFN production as the main mediator of pDCs' effect on humoral responses. A study in NHPs previously showed a correlation between type 1 IFN and spike-specific humoral responses induced by Ad26.COV2.S [12], and we have now directly demonstrated a mechanistic link with ChAdOx1 nCoV-19. Previous studies testing an adenovirus vector vaccine against foot and mouth disease virus (FMDV) in mice reported higher humoral and cellular responses when including porcine IFN-α in the construct [34,35]. Type 1 IFNs are known to support B cell activation and differentiation into plasma cells [36,37]. Previous studies also report the role of type 1 IFN in augmenting humoral responses to other vaccine modalities including LAIV and MVA [38–40], suggesting that this effect applies for humoral responses in general.

In addition to directly activating B cells, type 1 IFN also augmented Ad vector-induced humoral responses through promoting IL-6 secretion. In vivo studies in mice and hepatitis C patients receiving recombinant pegylated IFN-α injections show that type 1 IFN enhances production of IL-6 [41,42]. Historically named B-cell stimulatory factor-2, IL-6 is known to promote survival, expansion, and maturation of B cells to become plasmablasts [43]. This is supported by several clinical studies reporting decreased humoral responses to mRNA-based SARS-CoV-2 vaccines with anti-IL-6R treatment [44,45]. Interestingly, pDCs have been previously reported to produce both type I IFN and IL-6 and thereby modulate plasmablast differentiation [38]. However, in our system, we found that pDCs used the same cytokines but through a multi-step process that involved type I IFN signaling to cDCs as an intermediate cell type. Whether these differences reflect a distinct characteristic of Ad vector-induced immunity or a difference between the current lymphoid tissue-based organoid system compared to the previously used PBMC-based culture remains to be determined.

Type 1 IFN and IL-6 also modulate total CD4$^+$ T cell and T$_{FH}$ responses in our experimental model. Type 1 IFN promotes Bcl6 expression in CD4$^+$ T cells [46], but is inadequate to induce complete T$_{FH}$ differentiation. Full T$_{FH}$ commitment requires other cytokines, including IL-6 [47]. As type 1 IFN itself augments IL-6 secretion, it appears that type 1 IFN and IL-6 synergize to induce T$_{FH}$ differentiation and activation in response to adenovirus vaccination, in turn promoting T-dependent B cell responses. Other than a direct effect on T cells, type 1 IFN may also modulate cDCs to promote T$_{FH}$ differentiation [48]. Studies of adenoviral FMDV vaccines in mice reported that including type 1 IFN in the construct enhanced both induction and activation of T$_{FH}$ cells as well as the antibody production against the transgene [34,35].

Besides their cytokine production, pDCs may also contribute to humoral responses through producing target antigens. Given their efficient transduction by ChAdOx1, it is possible that pDCs can present antigens to T cells or provide antigens to B cells for the GC reaction. Several studies have reported that pDC subsets can contribute to antigen presentation to T cells, either by directly presenting [49,50] or by transferring antigens to cDCs [51,52]. Our data suggests that their production of type 1 IFN is their major mode of action in this model, given the near complete restoration of antibody production in pDC-depleted cultures by recombinant IFN-α supplementation. Nonetheless, the decrease in antibody production with type 1 IFN blockade was milder than pDC-depletion, which suggests that antigen production may still form a minor part of pDCs' role in the humoral response to adenovirus vectors.

One limitation of our study is the inability to completely map out the IL-6 pathway in this system. We discovered that a wide range of cell types produce IL-6 in response to ChAdOx1 nCoV-19, providing several possible sources of IL-6. Both B cells and cDCs pose potential important sources of ChAdOx1-induced IL-6 due to their abundance and high IL-6+ frequencies, respectively. However, we were unable to determine the relative contribution of B cells compared to cDCs (and other non-B cells) due to changes in relative cell abundance and possible alterations in cytokine production or usage in the selected/depleted cultures based on the experimental design. Alternate approaches will be required to definitively isolate the relative contribution of each cell type.

The tonsil organoid system provides an in vitro model for studying vaccine responses with cells derived from human SLO tissue that resembles lymph nodes closer than peripheral blood culture models. However, as specialized SLOs that are constantly exposed to pathogens in the oral mucosa, tonsils would still have differences with lymph nodes, and this unique biology may be particularly relevant when considering mucosal vaccination. The pathology prompting the removal of the tonsils may also affect the responses in the system, as evidenced by several donors that produced S1-specific antibody responses at baseline (and were thus excluded from analysis).

Using the in vitro tonsil organoid model, we were able to demonstrate humoral and cellular responses to ChAdOx1 nCoV-19. The system appears to model recall responses, which would be applicable to booster vaccinations. We discovered that pDCs modulate humoral responses to adenoviral vector vaccines, primarily through type 1 IFN. Type 1 IFN augmented IL-6 production, which also modulated humoral responses. Both type 1 IFN and IL-6 promoted ChAdOx1-induced CD4+ T and $T_{FH}$ responses, possibly partially linking cellular and humoral immunity. The system provides great potential for mechanistic studies on vaccine responses.

## Supporting information

**S1 Raw Data. Underlying data for all figures.**
(XLSX)

**S1 Fig. Gating scheme.** Gating strategy for **(A)** B cell, **(B)** innate, and **(C)** T cell panels.
(TIFF)

**S2 Fig. Additional data related to Fig 1. (A)** Experimental schematic of tonsil organoid culture system. Schematic was generated in BioRender (https://BioRender.com/3a5hazz). **(B)** Viability and cell recovery from tonsil organoid cultures over time. **(C)** Cumulative anti-S1 IgG production over time in donors with spontaneous anti-S1 IgG production. **(D)** Anti-S1 IgG production in tonsil organoids from pre-pandemic donors after 27 days of culture with or without ChAdOx1 nCoV-19 stimulation. **(E)** Cumulative total IgG production in post-pandemic tonsil organoids with or without ChAdOx1 nCoV-19 stimulation after 14-day culture. **(F)** Representative FACS plots of IgM and CD27 expression in B cell subsets from unstimulated and ChAdOx1 nCoV-19-stimulated organoids on day 7 of culture. **(G)** Representative FACS plot of B cells from unstimulated, ChAdOx1 nCoV-19-stimulated, and ChAdOx1 GFP-stimulated organoids on day 7 of culture. **(H)** Percentage of S1-specific B cells in unstimulated and ChAdOx1 nCoV-19-stimulated organoids from post-pandemic donors. Data in B are combined from 2 experiments with 9 donors, D from 1 experiment with 4 donors, E from 2 experiments with 8 donors, H from 4 experiments with 16 donors. Each symbol represents an individual donor. Values in D, E, and H were compared using Wilcoxon matched pairs signed rank test.
(TIFF)

**S3 Fig. Additional data related to Fig 2. (A)** Composition of cell types recovered after tonsil processing compared to after thawing frozen tonsil cells. **(B)** Representative FACS plots of ChAdOx1 transgene expression (either S1 spike or GFP) for each cell type. **(C)** Composition of cell types recovered from unstimulated and ChAdOx1 nCoV-19-stimulated tonsil organoids after 24 hours in culture. **(D)** Cytokine levels in media supernatants from unstimulated and ChAdOx1

nCoV-19-stimulated tonsil organoids after 24 hours in culture. Data in A are from 1 experiment with 4 donors, C from 3 experiments with a total of 12 donors, D from 4 experiments with a total of 16 donors. Each symbol represents an individual donor. Values in A were compared using multiple t-tests, values in C were compared using multiple Wilcoxon tests.
(TIFF)

**S4 Fig. Additional data related to Fig 3. (A)** Composition of cell types recovered from undepleted and pDC-depleted tonsil cells before plating. **(B)** Cumulative total IgG production in unstimulated or ChAdOx1 nCoV-19-stimulated organoids with or without pDC-depletion after 14-day culture. **(C)** Surface CD80 expression on different cell types in ChAdOx1 nCoV-19-stimulated tonsil organoids with or without pDC-depletion after 24 hours in culture. **(D)** Cytokine levels in media supernatants from ChAdOx1 nCoV-19-stimulated tonsil organoids with or without pDC-depletion after 24 hours in culture. Data in A are from 3 experiments with a total of 9 donors, B from 2 experiments with a total of 8 donors, C-D are combined from 4 experiments with a total of 14 donors. Each symbol represents an individual donor. Values in A were compared using multiple Wilcoxon tests, values in B were compared using Friedman test with Dunn's multiple comparisons test, values in C were compared using Wilcoxon matched pairs signed rank test.
(TIFF)

**S5 Fig. Additional data related to Fig 4. (A)** Transduction rates of different cell types in ChAdOx1 nCoV-19-stimulated tonsil organoids with or without type 1 IFN blockade after 24 hours in culture. **(B)** Levels of IFN-α, TNF, IFN-γ, and IL-6 in ChAdOx1 nCoV-19-stimulated tonsil organoids with or without type 1 IFN blockade after 24 hours in culture. **(C)** Cumulative anti-S1 IgG production in post-pandemic tonsil organoids with pDC-depletion and IFN-α supplementation over 14-day culture. **(D)** Percentage of total transduced cells after 24-hour culture in ChAdOx1 nCoV-19-stimulated tonsil organoids with or without pDC-depletion and IFN-α supplementation. **(E)** Cytokine levels in media supernatants from tonsil organoids stimulated with Ad5 nCoV-19 or ChAdOx1 nCoV-19 after 24 hours in culture. **(F)** Cumulative anti-S1 IgG production in tonsil organoids stimulated with Ad5 nCoV-19 with or without IFN-α supplementation or ChAdOx1 nCoV-19 after 14-day culture. Data in A-B are combined from 3 experiments with 10 donors, C and D from 3 experiments with 11 donors, E from 1 experiment with 8 donors, F from 1 experiment with 7 donors. Each symbol represents an individual donor. Values in A and B were compared using Wilcoxon matched pairs signed rank test, values in D-F were compared using Friedman test with Dunn's multiple comparisons test.
(TIFF)

**S6 Fig. Additional data related to Figs 6 and 7. (A)** Representative FACS plots of CD40L and intracellular cytokine (IFN-γ, TNF, and IL-2) expression in CD4+ T cells in ChAdOx1 nCoV-19-stimulated tonsil organoids. **(B-C)** Percentage of CD40L+CD69+ cells **(B)** and co-expressing CD40L and cytokines **(C)** of CD4+ T cells in tonsil organoids stimulated with ChAdOx1 nCoV-19 or ChAdOx1 GFP. **(D)** Representative FACS plots of CD69 and CD40L expression on T$_{FH}$ (CD4+PD-1+CXCR5+) cells in ChAdOx1 nCoV-19-stimulated tonsil organoids. All results were from cell harvest at 7 days after plating. Data in B and C are from 1 experiment with a total of 8 donors. Each symbol represents an individual donor. Values in B and C compared using Friedman test with Dunn's multiple comparisons test.
(TIFF)

## Acknowledgments

We thank the patients who consented to donate their tissue for the research, as well as the Oxford Radcliffe Biobank team for patient consenting and sample collection. We thank Dr. Helen Ferry for technical assistance in flow cytometry. We thank Prof. Dr. Marta Rizzi (University of Freiburg) for assistance in generating the SARS-CoV-2 spike peptide tetramers.

## Author contributions

**Conceptualization:** Nicholas M Provine.

**Data curation:** Maria Fransiska Pudjohartono.

**Formal analysis:** Maria Fransiska Pudjohartono.

**Funding acquisition:** Paul Klenerman, Nicholas M Provine.

**Investigation:** Maria Fransiska Pudjohartono.

**Methodology:** Maria Fransiska Pudjohartono, Nicholas M Provine.

**Resources:** Maria Fransiska Pudjohartono, Kate Powell, Eleanor Barnes.

**Supervision:** Eleanor Barnes, Paul Klenerman, Nicholas M Provine.

**Visualization:** Maria Fransiska Pudjohartono.

**Writing – original draft:** Maria Fransiska Pudjohartono.

**Writing – review & editing:** Maria Fransiska Pudjohartono, Kate Powell, Eleanor Barnes, Paul Klenerman, Nicholas M Provine.

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
