## [Decision Letter · Decision Letter 0]

9 Apr 2025

Human tonsil organoids reveal innate pathways modulating humoral and cellular responses to ChAdOx1

PLOS Pathogens

Dear Dr. Provine,

Thank you very much for submitting your manuscript "Human tonsil organoids reveal innate pathways modulating humoral and cellular responses to ChAdOx1" for review by PLOS Pathogens. Your manuscript was fully evaluated at the editorial level and by independent peer reviewers. The reviewers appreciated the attention to an important problem, but raised some substantial concerns about the manuscript as it currently stands. These issues must be addressed before we would be willing to consider a revised version of your study. We cannot, of course, promise publication at that time. We therefore ask you to modify the manuscript according to the review recommendations before we can consider your manuscript for acceptance. Your revisions should address the specific points made by each reviewer.

I am returning your manuscript with three reviews. The reviewers came to different conclusions about the paper, including one having significant concerns about the novelty of research findings. After reading the reviews and looking at the manuscript, I recommend Major Revision based on the critiques from the more critical reviews. I am sorry I cannot be more positive at the moment, however we are looking forward to receiving your revision.

Note that we may send your paper back to some of the more critical reviewers upon resubmission.

Please pay particular attention to the novelty concerns and major issues raised by reviewer 1 and 2 and give them due consideration.

(1) A letter containing a detailed list of your responses to the review comments and a description of the changes you have made in the manuscript.

(2) Two versions of the manuscript: one with either highlights or tracked changes denoting where the text has been changed; the other a clean version (uploaded as the manuscript file).

We hope to receive your revised manuscript within 60 days. If you anticipate any delay in its return, we ask that you let us know the expected resubmission date by replying to this email. Revised manuscripts received beyond 60 days may require evaluation and peer review similar to that applied to newly submitted manuscripts.

Kind regards,

Rahul Suryawanshi

Guest Editor

PLOS Pathogens

Sonja Best

Section Editor

Editor-in-Chief

PLOS Pathogens

orcid.org/0000-0003-2946-9497

Editor-in-Chief

PLOS Pathogens

orcid.org/0000-0002-7699-2064

**Journal Requirements:**

At this stage, the following Authors/Authors require contributions: Maria Fransiska Pudjohartono, Kate Powell, Eleanor Barnes, Paul Klenerman, and Nicholas M Provine. Please ensure that the full contributions of each author are acknowledged in the "Add/Edit/Remove Authors" section of our submission form.

https://journals.plos.org/plospathogens/s/submission-guidelines#loc-parts-of-a-submission

4) We do not publish any copyright or trademark symbols that usually accompany proprietary names, eg ©,  ®, or TM  (e.g. next to drug or reagent names). Therefore please remove all instances of trademark/copyright symbols throughout the text, including:

- TM on pages: 7, and 10.

5) Please upload all main figures as separate Figure files in .tif or .eps format. For more information about how to convert and format your figure files please see our guidelines: 

6) We have noticed that you have uploaded Supporting Information files, but you have not included a list of legends. Please add a full list of legends for your Supporting Information files after the references list.

7) Some material included in your submission may be copyrighted. According to PLOSu2019s copyright policy, authors who use figures or other material (e.g., graphics, clipart, maps) from another author or copyright holder must demonstrate or obtain permission to publish this material under the Creative Commons Attribution 4.0 International (CC BY 4.0) License used by PLOS journals. Please closely review the details of PLOSu2019s copyright requirements here: PLOS Licenses and Copyright. If you need to request permissions from a copyright holder, you may use PLOS's Copyright Content Permission form.

Potential Copyright Issues:

i) Figures 8, and S1A. Please confirm whether you drew the images / clip-art within the figure panels by hand. If you did not draw the images, please provide (a) a link to the source of the images or icons and their license / terms of use; or (b) written permission from the copyright holder to publish the images or icons under our CC BY 4.0 license. Alternatively, you may replace the images with open source alternatives. See these open source resources you may use to replace images / clip-art:

8) We note that your Data Availability Statement is currently as follows: "All relevant data are within the manuscript and its Supporting Information files." Please confirm at this time whether or not your submission contains all raw data required to replicate the results of your study. Authors must share the “minimal data set” for their submission. PLOS defines the minimal data set to consist of the data required to replicate all study findings reported in the article, as well as related metadata and methods (https://journals.plos.org/plosone/s/data-availability#loc-minimal-data-set-definition).

9) Please ensure that the funders and grant numbers match between the Financial Disclosure field and the Funding Information tab in your submission form. Note that the funders must be provided in the same order in both places as well. Currently, "Jardine Foundation Scholarship ,University of Oxford NDM COVID-19 emergency relief fun, and Pandemic Sciences Institute career fellowship" are missing from the Funding Information tab. The funders providing this grant (U19 I082360) are different in both places. In addition, the order of the funders is not the same in both locations.

**Reviewers' Comments:**

Reviewer's Responses to Questions

**Part I - Summary**

Reviewer #1: This manuscript describes the innate immune response to a chimpanzee adenovirus-vectored vaccine for SARS-CoV-2 (ChAdOx1 nCoV-19) using a human tonsil organoid model. The main conclusions of the paper are that AdV immunogenicity is associated with the innate activation of PDCs, which are the primary producers of T1 IFNs in the model. The authors also show that T1 IFN supports AdV responses by inducing IL-6 production, which can promote T follicular helper responses. For the most part, the experiments are rigorously designed and data presented are transparent. In my opinion, the findings represent an incremental advance to our existing understanding of the pDC/B cell axis and the roles of T1 IFN and IL-6 in these responses. T1 IFN responses to viral vectors are well described (https://doi.org/10.1016/j.ymthe.2020.01.001; https://doi.org/10.4049/jimmunol.179.3.1721;
https://doi.org/10.1172/JCI37607) and the role of pDCs in T1 IFN production in tonsil organoids is shown in the paper describing the model (https://doi.org/10.1038/s41591-020-01145-0). The downstream associated connections, that T1 IFNs support AdV responses through IL-6, which can promote CD4 T cell responses, has also been described previously (https://doi.org/10.1006/mthe.2002.0658 as an example). Given that the main take-homes have been described in other publications, I’m not convinced that sufficient novelty has been demonstrated.

Reviewer #2: In this article, Pudjohartono et al. work towards investigating the immune responses directed against ChAdOx1 using human tonsil organoids. The authors show that upon exposure of post- SARS-CoV2 pandemic tonsillar organoids, spike specific antibodies increase over time within tonsillar organoids, with an increase in spike specific antigen secreting B-cells. In addition to increase in several immune cell subsets, notably pDCs, there was also an increase in several cytokines such as IFN-a, TNF and IL-6, with decrease in cytokines such as IL-17a, IL-23 when compared to unstimulated controls. The increase in pDCs was shown to be crucial for the secretion of spike specific antibodies, with the IFN-a and IL-6 shown to augment B and Tfh cell responses.

The article is well written, succinct and the authors have clearly shown the important role of pDCs in generation of spike specific specific antibodies when exposed to ChAdOx1, with adenovirus based vaccines showing similar immune responses when used for other infections. There are a few concerns however that need to be addressed, namely the exact mechanism of how cytokines regulate antibody product

Reviewer #3: Pudjohartono et al utilized a human tonsil organoid model to study the regulation of adaptive responses to ChAdOx1 nCoV-19. They documented early innate immune activation and cytokine release and subsequent late T and B cell activation and antigen-specific antibody secretion. They further show that plasmacytoid dendritic cells (pDCs) are transduced the most and their IFN- α secretion was critical for humoral responses and IL-6 enhanced it. The authors present an interesting model to gain mechanistic insights into vaccines in vitro. The work is interesting and provides a new avenue to model vaccine research thereby warranting publication. Few minor points are suggested to improve the manuscript.

**Part II – Major Issues: Key Experiments Required for Acceptance**

Reviewer #1: I provide a few suggestions for strategies to improve the novelty of the work.

1) Antibody non-responders were discarded from the study early on. It would be potentially interesting to determine if their innate responses (T1 IFN and IL-6) were different than the responders, and if so, how.

2) One could imagine that different AdV might be more or less stimulatory of innate responses through T1 IFN and IL-6. Identifying vector-specific differences in the tuning for these cytokines could be of interest, as well as how those differences influence the down-stream B cell and T cell responses.

Reviewer #2: 1. It would be informative for the readers to provide the gating strategy (or strategies) used to define cell populations in the study (can provide this as a supplementary figure).

2. The lines 233-234, the authors state that there was higher CD27 expression when compared to other subsets. Is this comparison within the subsets (i.e. the other gated populations) of the ChAdOx1 treated cells or when compared to the unstimulated cells ? Based on Figure S1D (lower panel), it seems like CD27 expression is higher IgD-CD38++IgD- populations in unstimulated cells when compared to ChaAdOx1 treated cells. Please clarify.

3. While the amount of transduction rates are quite low, can the authors speculate as to why transduction rates in T-cells are significantly lowered when pDCs are depleted in ChAdOx1 treated samples (Figure 3D, T-cells) ?

4. Based on Figure 5, it is still unclear what cell produces IL-6 in the organoid model. Given the importance of IL-6 in generating S1 specific antibodies (Figure 5B), can the authors speculate what is the source of IL-6 in organoids ? The source of IL-6 is still something that is important to figure out. What is the reason for IL-6 secretion to increase when B-cells are depleted from the tonsillar organoids.

5. The authors clearly show the importance of IFNa and IL-6 in regulating immune responses towards ChAdOx1. In addition to these two cytokines, TNF is another cytokine that is found abundantly in ChAdOx1 treated samples and significantly affected by pDC depletion. Have the authors tried performing depletion of TNF to see its effect of antibody production and cellular activation ?

6. Figure 8 is a little confusing as a pathway of the immune responses and cells involved is not clear. Does IL-6 directly affect B-cell activation or does it work via. Tfh regulation ? What are the exact non B-cells that play a role in adenovirus based vaccine immunity ?

Reviewer #3: None

**Part III – Minor Issues: Editorial and Data Presentation Modifications**

Reviewer #1: Fig 3: pDC depletion: it is not a novel finding that pDCs, through T1 IFN and IL-6, are required to support a humoral response (Jego et al., Immunity 2003). It is also not a novel finding in the model used by the authors (Wagar et al., Nature Medicine 2021).

Fig 3/4: The authors show that SARS-CoV-2 specific antibodies are decreased when pDCs are depleted or T1 IFN is blocked. What about total (non-specific) antibody production?

Fig 5: The interpretation of these data are complicated by the possibility that cells under various selection and depletion conditions might have different preferences for using up the secreted cytokines (i.e., that there could be influences on both the secretion and the usage side).

Fig 6: The authors claim to measure spike-specific CD4 T cell responses. How do you know these are not simply responsive to the vector rather than spike protein?

Reviewer #2: (No Response)

Reviewer #3: Why do some donors don’t exhibit a response? The authors need to discuss this at least. Since the authors measured pre-existing S1-specific B cells, they should plot the antibody responses against those to demonstrate their point that “Our culture system appears to efficiently model a recall/boost response.”

Why did the vaccination suppressed unstimulated antibody production? The authors should at least discuss this in the text besides indicating that the samples will be excluded from the study.

Why would pre-pandemic not produce a response? The IgM data is interesting but do the authors have some metadata about the post-pandemic cohort? Are they previously infected? Vaccinated? A better characterization of the cohort, if possible, is necessary in the methods section. Critically, since the authors posit that pre-pandemic cultures do not have enough time to class switch, have the authors incubated them longer or provided repeat immunization to see if they stimulate antibody production?

PLOS authors have the option to publish the peer review history of their article (what does this mean? ). If published, this will include your full peer review and any attached files.

**Do you want your identity to be public for this peer review?** For information about this choice, including consent withdrawal, please see our Privacy Policy .

Reviewer #1: No

Reviewer #2: No

Reviewer #3: No

**Figure resubmission:**

**Reproducibility:**



---

## [Decision Letter · Decision Letter 1]

5 Aug 2025

Dear Provine,

We are pleased to inform you that your manuscript 'Human tonsil organoids reveal innate pathways modulating humoral and cellular responses to ChAdOx1' has been provisionally accepted for publication in PLOS Pathogens.

Best regards,

Rahul Suryawanshi

Guest Editor

PLOS Pathogens

Sonja Best

Section Editor

PLOS Pathogens

Sumita Bhaduri-McIntosh

Editor-in-Chief

PLOS Pathogens

orcid.org/0000-0003-2946-9497

Michael Malim

Editor-in-Chief

PLOS Pathogens

orcid.org/0000-0002-7699-2064

Reviewer Comments (if any, and for reference):

Reviewer's Responses to Questions

**Part I - Summary**

Reviewer #2: (No Response)

Reviewer #3: The authors have addressed all my concerns.

**Part II – Major Issues: Key Experiments Required for Acceptance**

Reviewer #2: N/A

Reviewer #3: (No Response)

**Part III – Minor Issues: Editorial and Data Presentation Modifications**

Reviewer #2: N/A

Reviewer #3: (No Response)

PLOS authors have the option to publish the peer review history of their article (what does this mean? ). If published, this will include your full peer review and any attached files.

**Do you want your identity to be public for this peer review?** For information about this choice, including consent withdrawal, please see our Privacy Policy .

Reviewer #2: No

Reviewer #3: No

---

## [Editor Report · Acceptance letter]

Dear Provine,

We are delighted to inform you that your manuscript, "Human tonsil organoids reveal innate pathways modulating humoral and cellular responses to ChAdOx1," has been formally accepted for publication in PLOS Pathogens.

Best regards,

Sumita Bhaduri-McIntosh

Editor-in-Chief

PLOS Pathogens

orcid.org/0000-0003-2946-9497

Michael Malim

Editor-in-Chief

PLOS Pathogens

orcid.org/0000-0002-7699-2064